# Morphometric geometric differences between right and left human tali: A cadaveric study of fluctuating asymmetry via systematic measurement and three-dimensional scanning

Chayanin Angthong[1]*, Prasit Rajbhandari[2], Andrea Veljkovic[3], Atthaporn Piyaphanee[4], Sjoerd Antoine Sebastian Stufkens[5], Ricky Wibowo[6]

1 Department of Orthopaedics, Faculty of Medicine, Thammasat University, Pathum Thani, Thailand, 2 Department of Orthopaedic Surgery, Manmohan Memorial Medical College and Teaching Hospital, Kathmandu, Nepal, 3 Department of Orthopaedics, Foot and Ankle Reconstruction/Arthroscopy & Athletic Injuries Knee and Ankle/Trauma, St. Paul's Hospital, The University of British Columbia, Footbridge Clinic, Vancouver, BC, Canada, 4 Department of Orthopaedic Surgery, Phra Nangklao Hospital, Nonthaburi, Thailand, 5 Department of Orthopaedic Surgery, Academic Medical Center, University of Amsterdam, Amsterdam, The Netherlands, 6 Department of Orthopaedics and Traumatology, Faculty of Medicine, Padjajaran University/Dr. Hasan Sadikin Hospital Bandung, Bandung, Indonesia

* chatthara@yahoo.com

## Abstract

### Background

Little is known about differences in the size and morphology of the right and left human tali. The present study demonstrates differences between right and left talar morphometric geometric profiles as fluctuating asymmetry in matched pairs of cadaveric specimens.

### Methods

In total, 24 tali were collected in this study. All eligible tali were systematically measured with a Vernier caliper and three-dimensional laser scanner, which provided data for further analysis regarding the talar morphometric geometric profiles. Data were calculated to demonstrate differences between the right and left talar profiles using a matched-pair method, including the general size of the talus.

### Results

The average talar length was 53.5 mm, the average talar dome height was 31.2 mm, and the average talar body width was 41.3 mm. The average anterior trochlear width, middle trochlear width, posterior trochlear width, and trochlear length were 31.8, 31.2, 28.3, and 30.7 mm, respectively. Eleven matched pairs of intact tali were eligible for the matched-pair study. Paired t-tests showed significant differences in the talar dome height (P = 0.019), middle trochlear width (P = 0.027), and posterior trochlear width (P = 0.016) between the right

**Data Availability Statement:** All relevant data are within the paper and its Supporting Information files.

**Funding:** Dr. Angthong reports grants from Thammasat University (grant number: 2/43/ 2561, year 2018), personal fees from Phoenix Surgical Equipment (Thailand) Co., Ltd, personal fees from Amgen, personal fees from THAI ADK ENGINEERING COMPANY LIMITED, during the conduct of the study; In addition, Dr. Angthong has a patent Manufacture protocol for the artificial total talus (Thailand patent number: 14879) issued.

**Competing interests:** Dr. Angthong reports grants from Thammasat University (grant number: 2/43/ 2561, year 2018), personal fees from Phoenix Surgical Equipment (Thailand) Co., Ltd, personal fees from Amgen, personal fees from THAI ADK ENGINEERING COMPANY LIMITED, during the conduct of the study; In addition, Dr. Angthong has a patent Manufacture protocol for the artificial total talus (Thailand patent number: 14879) issued. This does not alter our adherence to PLOS ONE policies on sharing data and materials.

and left tali. However, there were no significant differences in the surface area or volume between the right and left tali.

## Conclusion

Significant differences in the morphometric profile were found between the right and left matched pairs of tali. This basic information indicates that the profile of the contralateral talus may not be used as a single reference to reconstruct or duplicate the talus of interest in certain conditions such as talar prosthesis implantation or customized total ankle replacement.

## Introduction

Knowledge and technology in the field of foot and ankle surgery have progressed rapidly in recent decades. This is evident by the advances in total ankle replacement surgery [1], replacement surgery using total artificial tali [2], and arthroscopic surgery [3]. Such substantial technological progress in foot and ankle surgery has been made possible by the knowledge gained from morphological research in technologies such as computed tomography (CT) [4], three-dimensional (3D) printing technology [5], and computer-aided design programs, which transform CT images into 3D images [2]. These advanced technologies have led to developments in cutting-edge orthopedic surgery, such as the successful use of patient-specific 3D-printed titanium implants for the treatment of complex large bony defects and deformities and for use in arthrodesis procedures [5]. Prostheses have also been created to replace damaged or missing original bone [2], [6]. One of these advances is the development of the total talar prosthesis or customized total ankle prosthesis. The size and shape of the total talar prosthesis is designed based on the contralateral talus [2]. According to Zaidi, human bodies show both anatomical and functional asymmetries, some of which are of great clinical significance. Thus, the study of human body asymmetry is important to achieve an accurate diagnosis, perform effective treatment, and establish a management plan [7]. However, little is known about the anatomical and morphological variations of the foot and ankle in humans, especially in a pivotal bone such as the talus. Essential data needed for the manufacturing process and the outcomes of talar replacement surgery are related to variations in size, volume, and surface area between the right and left tali. The present study was performed to demonstrate the morphometric geometric variations between the right and left tali in matched pairs of human cadaveric specimens. The authors hypothesized that fluctuating asymmetry [8] [9] involving morphometric geometric differences exists between the bilateral tali and that this asymmetry may alter the current method used to manufacture templates from the normal contralateral talus.

## Material and methods

Twenty-four tali from 12 matched pairs of fresh human cadaveric ankles were collected in this study. All tali were removed from the tibiotalar, talocalcaneal, and talonavicular joints with an effort to preserve the native cartilage by trained research assistants (resident physicians/fellows) under the supervision of fellowship-trained foot and ankle surgeons. All tali were systematically measured with a Vernier caliper to document their morphometric profiles using the method established by the senior author and Eun et al. (Figs 1–3) [10]. Each talar measurement was performed by the research assistants (resident physicians/fellows) under supervision

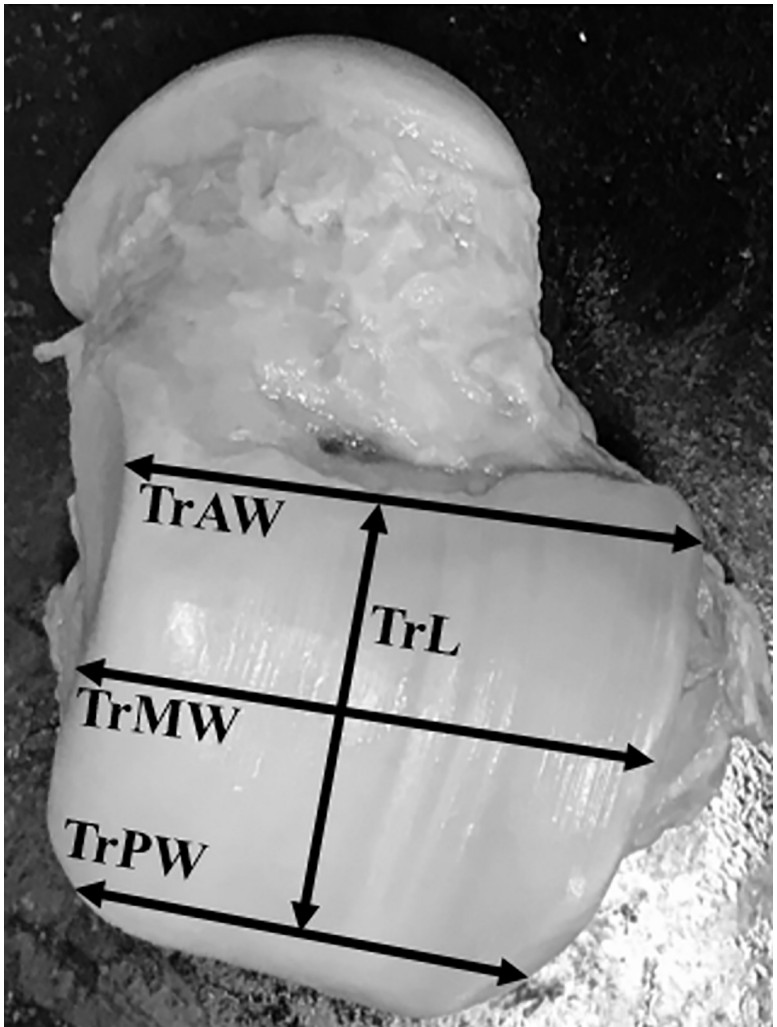

**Fig 1. Abbreviations: Talar length (TaL); Anterior trochlear width (TrAW); Middle trochlear width (TrMW); Posterior trochlear width (TrPW); Trochlear length (TrL).**

by a trained research fellow. All tali were also scanned using a 3D laser scanner (HandySCAN 700; Creaform, Levis, Canada) to create a database of geometric profiles by a technician. These methods provided essential data for further analyses of morphologic variances between the right and left tali. A computer program (Autodesk Netfabb version 2020; Autodesk, Inc., San Rafael, CA, USA) was used to analyze the surface area and volume of each talus (Fig 4). The data were analyzed to assess the difference in measurement parameters, including the general size of the talus, between the right and left tali using a matched-pair method. Parallax error was avoided by keeping the eye in a straight line directly above the marking of the Vernier caliper. One talus was excluded because of morphologic changes caused by crystal deposition from an underlying disease affecting the donor cadaver. The remaining tali showed no signs of trauma, arthritis, previous hindfoot surgery, gouty arthritis, or other abnormalities. Therefore, both above-mentioned matched tali were excluded from the matched-pair study, and data were retrieved from 11 matched pairs of tali. In addition, baseline data of each cadaver were collected (including age, sex, and height) and recorded in the database for further analyses. The present study was approved by the ethics committee of the authors' institution (The

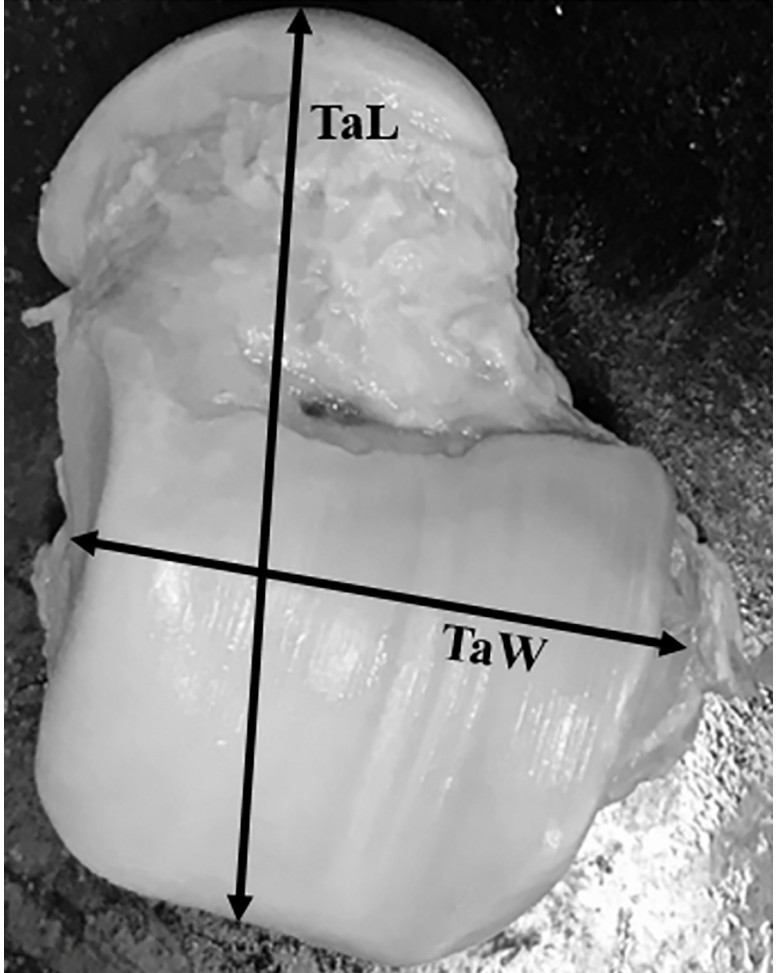

**Fig 2. Abbreviations: Talar length (TaL); Talar body width (TaW).**

Human Research Ethics Committee of Thammasat University No.1 (Faculty of Medicine)). The author's institute had discussed with the patients who provided the informed written consent prior to death for the use of their bodies in the education of medicine and medical research.

Statistical analysis was performed using IBM SPSS software version 22 (IBM Corp., Armonk, NY, USA). Differences in quantitative data were analyzed using Student's t-test because the data were parametric. Pearson's correlation coefficient was used to express the correlation between the parametric parameters. A P value of $<0.05$ was considered statistically significant.

## Results

In total, 24 tali from 12 matched-pair ankles were collected from 10 male and 2 female cadavers. The mean age of the cadavers was 69.4 ± 11.6 years. The mean height of the cadavers was 163.2 ± 5.6 cm. With respect to the morphometric profiles, an average talar length was 53.5 mm, the average talar dome height was 31.2 mm, and the average talar body width was 41.3 mm. The average anterior trochlear width, middle trochlear width, posterior trochlear width, and trochlear length were 31.8, 31.2, 28.3, and 30.7 mm, respectively. Paired t-tests among the

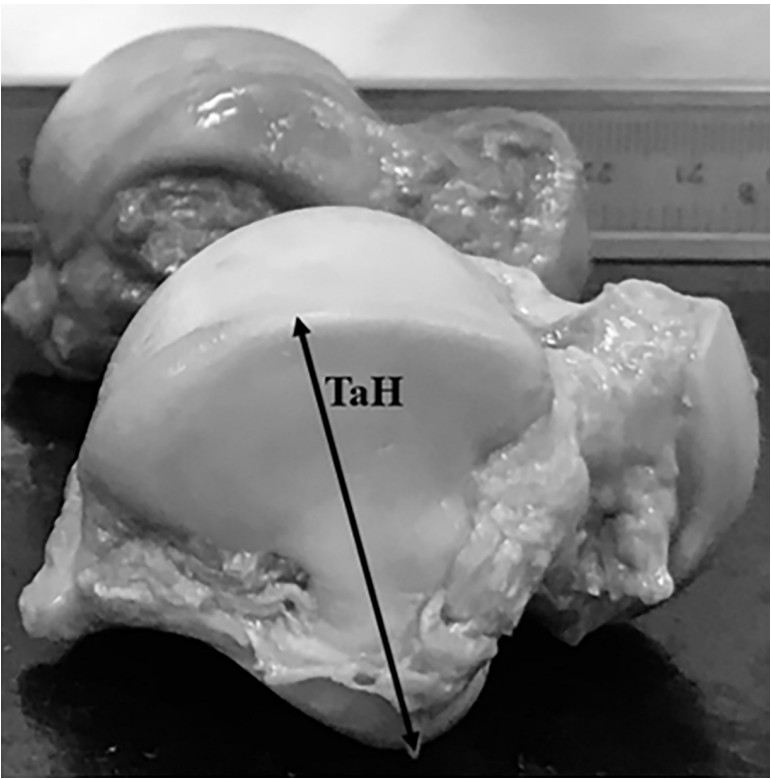

**Fig 3. Abbreviations: Talar dome height (TaH).**

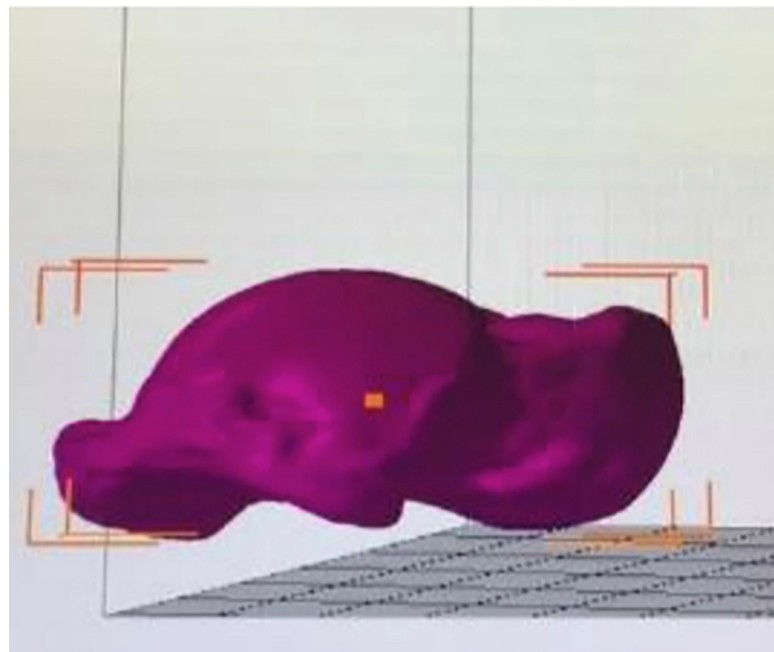

**Fig 4. The image of scanned talus from the computer program (Autodesk Netfabb version 2020, USA) which was used to analyze the surface area and volume of each talus.**

**Table 1. The morphometric profiles of talus from 11 matched pairs of tali.**

| | | Mean | Number | Std. Deviation | Std. Error Mean |
|---|---|---|---|---|---|
| Pair 1 | TaL_R | 53.95 | 11 | 3.08 | .93 |
| | TaL_L | 53.13 | 11 | 4.30 | 1.29 |
| Pair 2 | TaH_R | 32.18 | 11 | 2.44 | .73 |
| | TaH_L | 30.09 | 11 | 2.90 | .87 |
| Pair 3 | TaW_R | 40.86 | 11 | 2.47 | .74 |
| | TaW_L | 41.81 | 11 | 2.92 | .88 |
| Pair 4 | TrAW_R | 30.45 | 11 | 3.71 | 1.12 |
| | TrAW_L | 33.27 | 11 | 6.64 | 2.00 |
| Pair 5 | TrMW_R | 33.09 | 11 | 2.94 | .88 |
| | TrMW_L | 29.45 | 11 | 5.76 | 1.73 |
| Pair 6 | TrPW_R | 30.93 | 11 | 3.22 | .97 |
| | TrPW_L | 25.81 | 11 | 5.36 | 1.61 |
| Pair 7 | TrL_R | 32.54 | 11 | 3.53 | 1.06 |
| | TrL_L | 28.59 | 11 | 8.13 | 2.45 |

Abbreviations: Talar length (TaL); Talar dome height (TaH); Talar body width (TaW); Anterior trochlear width (TrAW); Middle trochlear width (TrMW); Posterior trochlear width (TrPW); Trochlear length (TrL); Standard (Std).

11 matched pairs of tali showed statistically significant differences in the talar dome height (P = 0.019), middle trochlear width (P = 0.027), and posterior trochlear width (P = 0.016) between the right and left tali (Tables 1 and 2).

Based on the 3D analysis of geometric profiles using the computer program (Autodesk Netfabb version 2020), the surface area and volume of the tali were calculated and are listed in Table 3. There were no significant differences in the surface area or volume between the right and left tali (Table 4).

## Discussion

The present study highlights the morphometric geometric variations between the right and left tali in matched pairs of fresh human cadaveric ankles. The results demonstrated significant differences between the left and right matched-pair talar morphometric profiles, specifically in

**Table 2. The morphometric differences of talus from 11 matched pairs of tali.**

| | | Paired Differences | | | | | Sig. (2-tailed) |
|---|---|---|---|---|---|---|---|
| | | Mean | Std. Deviation | Std. Error Mean | 95% Confidence Interval of the Difference | | |
| | | | | | Lower | Upper | |
| Pair 1 | TaL_R—TaL_L | .81 | 5.79 | 1.74 | -3.07 | 4.71 | .65 |
| Pair 2 | TaH_R—TaH_L | 2.09 | 2.47 | .74 | .42 | 3.75 | .01* |
| Pair 3 | TaW_R—TaW_L | -.95 | 4.17 | 1.25 | -3.75 | 1.84 | .46 |
| Pair 4 | TrAW_R—TrAW_L | -2.81 | 8.56 | 2.58 | -8.57 | 2.93 | .30 |
| Pair 5 | TrMW_R—TrMW_L | 3.63 | 4.65 | 1.40 | .51 | 6.76 | .02* |
| Pair 6 | TrPW_R—TrPW_L | 5.11 | 5.86 | 1.76 | 1.17 | 9.05 | .01* |
| Pair 7 | TrL_R—TrL_L | 3.95 | 6.73 | 2.02 | -.56 | 8.47 | .08 |

*Significant difference. Abbreviations: Talar length (TaL); Talar dome height (TaH); Talar body width (TaW); Anterior trochlear width (TrAW); Middle trochlear width (TrMW); Posterior trochlear width (TrPW); Trochlear length (TrL); Standard (Std); Significance (Sig).

**Table 3. The geometric profiles of talus from 11 matched pairs of tali.**

|  |  | Mean | Number | Std. Deviation | Std. Error Mean |
|---|---|---|---|---|---|
| Pair 1 | Talar surface (Right) | 68.59 | 11 | 5.47 | 1.65 |
|  | Talar surface (Left) | 68.06 | 11 | 6.06 | 1.82 |
| Pair 2 | Talar volume (Right) | 36.64 | 7 | 3.02 | 1.14 |
|  | Talar volume (Left) | 35.58 | 7 | 2.45 | .92 |

Standard (Std).

three important parameters. However, there were no significant differences in the surface area or volume between the right and left tali.

The talus is a pivotal bone in the foot and ankle. Its key functions are related to both motion preservation of the ankle and load distribution [11]. With respect to these unique functions, the native shape of the talus is also unique and asymmetrical because of its three articulations with the tibia, fibula, navicular bone, and calcaneus. Several ligaments attach the talus to its surrounding articular structures [12]. Based on these exclusive characteristics, the talus is prone to several pathologies and injuries, but the native talus is difficult to reconstruct or duplicate in patients with conditions such as peritalar instability [13], talar bone damage [14], and loss of the talus [2]. Several reports have proposed the characteristics of the talus in terms of its biomechanical and morphologic aspects. Akiyama et al. reported that the dorsal articular side of the talocrural joint could withstand fewer loads than the ventral side [4]. Angthong reported the trochlear length/width ratio of the talus from a 3D file study based on CT images [2]. These data were inconsistent with the data from previous studies by Fessy et al [15]., Stagni et al. [16]. and Kuo et al [17]. Islam et al. conducted a study on the morphology of the right and left tali in the same living subjects using a CT database with contributions by 3D analyses [18]. They found a strong degree of symmetry between the two sides [18]. However, their measurements were derived from simulations made from CT images [18]. Based on measurement of the actual talus with no need for simulation, the present study combined data collection via systematic Vernier caliper measurement of morphometric profiles and 3D scans using a 3D scanner to obtain geometric measurement parameters of the human talus. Direct combination of these measurement methods revealed morphologic variations in contrast to the findings by Islam et al. The right and left tali demonstrated several differences in terms of their morphometric profiles. These deviations from bilateral symmetry may exceed the body's ability to maintain its usual development in stressful conditions [8, 9]. Regarding the stressful condition, Julius Wolff and others realized that mechanical loads can affect bone architecture in living beings [19]. The fluctuating asymmetry may occur from this mechanism. In a meta-anaysis done by Beasley et al they found that fluctuating asymmetry is a sensitive biomarker of environmental stress. They concluded that fluctuating asymmetry as a biomarker of environmental

**Table 4. Differences of geometric profiles from 11 matched pairs of tali.**

|  |  | Paired Differences | | | | | Sig. (2-tailed) |
|---|---|---|---|---|---|---|---|
|  |  | Mean | Std. Deviation | Std. Error Mean | 95% Confidence Interval of the Difference | | |
|  |  |  |  |  | Lower | Upper |  |
| Pair 1 | Talar surface (Right versus Left) | .52 | 2.31 | .69 | -1.03 | 2.08 | .46 |
| Pair 2 | Talar volume (Right versus Left) | 1.05 | 1.83 | .69 | -.64 | 2.75 | .18 |

Standard (Std); Significance (Sig).

stress is a legitimate tool particularly when studies verify the biological relevance of stressors for the study organism [20]. Bone asymmetry is thought to basically result from disproportionate mechanical stress which influences bone remodeling and plasticity [21, 22]. Similarly, Hallgrímsson found evidence suggesting an increase in fluctuating asymmetry of skeletal traits with increasing age in a heterogeneous sample [23]. Other studies also have found that fluctuating asymmetry is highest at the extremes of age i.e elderly people have more body asymmetry [24, 25]. Our study found the consistent findings in accordance with the previous studies. These evidences recommended that the talar-related implant may need to customize its size regarding the specific side of the ankle especially on the elderly patient. Angthong found that the mean age of patients who had the indications for the total ankle replacement was 65.2 years [26]. This age number was also high and seemed to be closed to the mean age of cadavers (69.4 years) in the present study. The present study's evidence may help support the concept of patient-specific implant in the total ankle replacement for the elderly patient. In a study done by Teo et al they found 89.13% of right handed participants were also right legged. Right handed individuals are 8.2 times more likely to have ipsilateral leg dominance [27]. Shugaba et al also found similar result with 85% right handed participants were also right footed [28]. They found that the dominant leg was associated with greater volume which is consistent with our findings where the size of the right talus was more than the left ones.

In addition, the deviations in these parameters are important in terms of talar reconstruction/surgery and the manufacture of total talar prostheses for duplication and replacement of this bone. These results also support the emphasis on the manufacture of talar components of patient-specific size for use in total ankle arthroplasty implants [29]. Even if the measurement of the ankle-related bones had been done for the sizing of implant in the total ankle arthroplasty, Hsu et al found that talar implant sizing was not as accurate due to individual surgeon preference regarding the extent of gutter debridement [30]. Based on the results of the present study, measurement of the contralateral talus may not be the only useful method for determining the size of the talus of interest. Future 3D measurement of the talus on the pathologic/injured side may require the use of mirrored reconstruction images from articulations of the surrounding bones, such as the tibial plafond, navicular bone, and calcaneal facets to develop a more anatomically morphometric geometric template for further reconstruction or manufacturing than the conventionally manual measurement on two dimensional images. Similarly, CT images are extremely consistent and can be easily reconstructed to 3D model through computer software [31]. It is also important to do measurement of talar dome of the same side when aiming to design more anatomical ankle implant [32]. This technically challenging issue requires further study.

The main limitation of the present study is that the sample size was limited to 11 pairs of cadaveric ankles. However, the results still showed significant differences in the morphometric profiles between the right and left tali. Further study involving a larger sample size is needed to validate this information in a larger population. Regarding other limitation, the present study had no available data about the occupation and weight of the cadavers before their deaths. In addition to age, gender and height of cadavers, the bone morphometry might have the influence from these factors which should be considered in the further study. However, this study conducted the matched-pair analysis as right versus left ankle in the same cadaver; therefore, the different occupation and/or weight among cadavers may have less effect to the results of analysis.

## Conclusion

The present study confirmed significant differences in the morphometric profiles as fluctuating asymmetry in the bilateral tali, providing basic information for talar-related treatment.

The morphometric profile of the contralateral talus may not be used as a single reference to reconstruct or duplicate the talus of interest in certain conditions as discussed above.

## Supporting information

**S1 Data. The supplementary file of the data of this study (Microsoft excel form) was uploaded in the submission system.**
(XLSX)

## Acknowledgments

We also thank Angela Morben, DVM, ELS, from Edanz Group (https://en-author-services.edanzgroup.com/), for editing a draft of this manuscript.

## Author Contributions

**Conceptualization:** Chayanin Angthong, Sjoerd Antoine Sebastian Stufkens.

**Data curation:** Chayanin Angthong, Prasit Rajbhandari.

**Formal analysis:** Chayanin Angthong, Ricky Wibowo.

**Funding acquisition:** Chayanin Angthong.

**Investigation:** Chayanin Angthong, Prasit Rajbhandari, Atthaporn Piyaphanee, Ricky Wibowo.

**Methodology:** Chayanin Angthong, Prasit Rajbhandari.

**Project administration:** Chayanin Angthong.

**Software:** Chayanin Angthong.

**Supervision:** Chayanin Angthong, Prasit Rajbhandari.

**Validation:** Chayanin Angthong.

**Writing – original draft:** Chayanin Angthong, Prasit Rajbhandari.

**Writing – review & editing:** Chayanin Angthong, Prasit Rajbhandari, Andrea Veljkovic, Sjoerd Antoine Sebastian Stufkens.

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
