## [Decision Letter · Decision Letter 0]

19 Feb 2020

PONE-D-19-33551

Morphometric geometric differences between right and left human tali: A cadaveric study of fluctuating asymmetry via systematic measurement and three-dimensional scanning

PLOS ONE

Dear Dr. Angthong,

Thank you for submitting your manuscript to PLOS ONE. After careful consideration, we feel that it has merit but does not fully meet PLOS ONE’s publication criteria as it currently stands. Therefore, we invite you to submit a revised version of the manuscript that addresses the points raised during the review process.

We would appreciate receiving your revised manuscript by Apr 04 2020 11:59PM. To enhance the reproducibility of your results, we recommend that if applicable you deposit your laboratory protocols in protocols.io, where a protocol can be assigned its own identifier (DOI) such that it can be cited independently in the future. For instructions see: http://journals.plos.org/plosone/s/submission-guidelines#loc-laboratory-protocols

We look forward to receiving your revised manuscript.

Kind regards,

Shang-Chun Guo

Academic Editor

PLOS ONE

Journal Requirements:

1. In the ethics statement in the manuscript and in the online submission form, please provide additional information about the human tissues used in this study. Specifically, please ensure that you have discussed whether next-of-kin provided informed written consent for the use of the tissues. If patients provided informed written consent prior to death to have their bodies used in medical research, please include this information.

2. Thank you for including your ethics statement:  'This study was approved by the ethical committee of the institution's author.'

3. Thank you for including the following funding information within the acknowledgements section of your manuscript; "The author would like to propose the special thanks to Thammasat University that provide the research grant for this research project (grant number: ทป2/43/ 2561, year 2018)"

"Dr. Angthong reports grants from Thammasat University, during the conduct of the study; other from Amgen, personal fees from Phoenix Surgical Equipment (Thailand) Co., Ltd, outside the submitted work; In addition, Dr. Angthong has a patent Manufacture protocol for the artificial total talus issued"

4. a. Thank you for including your funding statement; "Dr. Angthong reports grants from Thammasat University, during the conduct of the study; other from Amgen, personal fees from Phoenix Surgical Equipment (Thailand) Co., Ltd, outside the submitted work; In addition, Dr. Angthong has a patent Manufacture protocol for the artificial total talus issued"

We note that you received funding from a commercial source: AMgen and Phoenix Surgical Equipment (Thailand) Co., Ltd

b.

We note that you have a patent relating to material pertinent to this article. Please provide an amended statement of Competing Interests to declare this patent (with details including name and number), along with any other relevant declarations relating to employment, consultancy, patents, products in development or modified products etc. Please confirm that this does not alter your adherence to all PLOS ONE policies on sharing data and materials, as detailed online in our guide for authors http://journals.plos.org/plosone/s/competing-interests by including the following statement: "This does not alter our adherence to  PLOS ONE policies on sharing data and materials.” If there are restrictions on sharing of data and/or materials, please state these. Please note that we cannot proceed with consideration of your article until this information has been declared.

5. Please amend your supporting information COI files to 'other' as these need not be included within your published manuscript

Reviewers' comments:

Reviewer's Responses to Questions

**Comments to the Author**

1. Is the manuscript technically sound, and do the data support the conclusions?

Reviewer #1: Yes

Reviewer #2: Partly

2. Has the statistical analysis been performed appropriately and rigorously? 

Reviewer #1: Yes

Reviewer #2: Yes

3. Have the authors made all data underlying the findings in their manuscript fully available?

Reviewer #1: Yes

Reviewer #2: No

4. Is the manuscript presented in an intelligible fashion and written in standard English?

Reviewer #1: Yes

Reviewer #2: Yes

5. Review Comments to the Author

Reviewer #1: In this study, a total of 24 tali were measured with a vernier caliper and three-dimensional laser scanner. The significant differences in the morphometric profile were found between the right and left matched pairs of tali. However, there are some points that need to be addressed.

1. Is it possible that the differences in the morphometric profile between the right and left matched pairs of tali due to the samples of elderly cadavers？

2. Please use a unified decimal place in the tables.

Reviewer #2: In this study, the authors demonstrated the morphometric geometric variations between the right and left tali in matched pairs of human cadaveric specimens. There are a few concerns on the methodology and interpretation of the findings.

1. In addition to age, gender and height, we understand that the bone morphometry especially for calcaneum and talus, also depends on the occupation and weight. Did the authors consider these factors as well?

2. From line 134 to line 136, the authors mentioned that 'Differences in categorical data were analyzed using the chi-square … Pearson's correlation coefficient was used to express the correlation between the parametric parameters'. In the Result section, I couldn't find the findings using correlation analyses or analyses of categorical data. Is this part of analysis missing in the Result section?

3. From line 201 to line 207, the authors cited several articles using different methods to measure the talus morphometry. However, they didn't describe whether their findings are similar/different from the others. What is the purpose of citing these articles?

4. It is not clear what 'stressful conditions' the authors are referring to in line 213.

5. In Table 1, five out of seven parameters of tali showed that right tali were larger than left tali. It seems that there is right dominance of talus morphometry, like handedness. The authors did not discuss the potential relationship between limb dominance and talus morphometry.

6. The clinical significance is not clear. Ankle arthroplasty implant definitely requires the measurements on the articulations of the bones surrounding the pathologic/injured talus. It is not clear how the findings in this study contribute to the improvement of current technology of making the implants.

6. PLOS authors have the option to publish the peer review history of their article (what does this mean?). If published, this will include your full peer review and any attached files.

Reviewer #1: No

Reviewer #2: No

---

## [Author Response · Author response to Decision Letter 0]

18 Mar 2020

We have responded to the editor and reviewers' comments as point to point as in the attached letter of revision. Thank you for your consideration.

---

## [Decision Letter · Decision Letter 1]

7 Apr 2020

Morphometric geometric differences between right and left human tali: A cadaveric study of fluctuating asymmetry via systematic measurement and three-dimensional scanning

PONE-D-19-33551R1

Dear Dr. Angthong,

We are pleased to inform you that your manuscript has been judged scientifically suitable for publication and will be formally accepted for publication once it complies with all outstanding technical requirements.

With kind regards,

Shang-Chun Guo

Academic Editor

PLOS ONE

Additional Editor Comments (optional):

Reviewers' comments:

Reviewer's Responses to Questions

**Comments to the Author**

1. If the authors have adequately addressed your comments raised in a previous round of review and you feel that this manuscript is now acceptable for publication, you may indicate that here to bypass the “Comments to the Author” section, enter your conflict of interest statement in the “Confidential to Editor” section, and submit your "Accept" recommendation.

Reviewer #1: All comments have been addressed

Reviewer #2: All comments have been addressed

2. Is the manuscript technically sound, and do the data support the conclusions?

Reviewer #1: Yes

Reviewer #2: (No Response)

3. Has the statistical analysis been performed appropriately and rigorously? 

Reviewer #1: Yes

Reviewer #2: (No Response)

4. Have the authors made all data underlying the findings in their manuscript fully available?

Reviewer #1: Yes

Reviewer #2: (No Response)

5. Is the manuscript presented in an intelligible fashion and written in standard English?

Reviewer #1: Yes

Reviewer #2: (No Response)

6. Review Comments to the Author

Reviewer #1: In this study, a total of 24 tali were measured with a vernier caliper and three-dimensional laser scanner. The significant differences in the morphometric profile were found between the right and left matched pairs of tali.

Thanks for your revision.

Reviewer #2: Minor suggestions: In Tables 1-4, the units for the numbers under the columns of Mean, SD and SEM are missing.

7. PLOS authors have the option to publish the peer review history of their article (what does this mean?). If published, this will include your full peer review and any attached files.

Reviewer #1: No

Reviewer #2: No

---

## [Editor Report · Acceptance letter]

10 Apr 2020

PONE-D-19-33551R1 

Morphometric geometric differences between right and left human tali: A cadaveric study of fluctuating asymmetry via systematic measurement and three-dimensional scanning 

Dear Dr. Angthong:

I am pleased to inform you that your manuscript has been deemed suitable for publication in PLOS ONE. Congratulations! Your manuscript is now with our production department. 

With kind regards,

on behalf of

Dr. Shang-Chun Guo 

Academic Editor

PLOS ONE